# Deep Learning for Automated Elective Lymph Node Level Segmentation for Head and Neck Cancer Radiotherapy

**DOI:** 10.3390/cancers14225501

**Published:** 2022-11-09

**Authors:** Victor I. J. Strijbis, Max Dahele, Oliver J. Gurney-Champion, Gerrit J. Blom, Marije R. Vergeer, Berend J. Slotman, Wilko F. A. R. Verbakel

**Affiliations:** 1Department of Radiation Oncology, Amsterdam UMC Location Vrije Universiteit Amsterdam, De Boelelaan 1117, 1081 HV Amsterdam, The Netherlands; 2Cancer Center Amsterdam, Cancer Treatment and Quality of Life, 1081 HV Amsterdam, The Netherlands; 3Department of Radiology and Nuclear Medicine, Amsterdam UMC Location University of Amsterdam, Meibergdreef 9, 1105 AZ Amsterdam, The Netherlands; 4Cancer Center Amsterdam, Imaging and Biomarkers, 1081 HV Amsterdam, The Netherlands

**Keywords:** computed tomography, deep learning, head-and-neck cancer, lymph nodes, radiation oncology, auto-contouring

## Abstract

**Simple Summary:**

When treating patients with head-and-neck cancer (HNC), in addition to the primary tumour, commonly involved lymph node (LN) levels are often electively irradiated. This requires the definition of the elective LN target volume. Because the LN levels that will be included in the target depend on the clinical situation, and because manual contouring is a laborious task that can also introduce inter- and intra-observer variation, being able to automate the segmentation of individual LN levels would reduce the clinical burden and would allow use of contours regardless of the primary tumor location. We trained and evaluated three patch- and/or voxel-based deep learning frameworks to segment elective LN levels. Our results suggest that accurate segmentations can be obtained using an ensemble of patch-based UNets and that this result can be further refined by sequentially applying a 2.5D, multi-view voxel classification network.

**Abstract:**

Depending on the clinical situation, different combinations of lymph node (LN) levels define the elective LN target volume in head-and-neck cancer (HNC) radiotherapy. The accurate auto-contouring of individual LN levels could reduce the burden and variability of manual segmentation and be used regardless of the primary tumor location. We evaluated three deep learning approaches for the segmenting individual LN levels I–V, which were manually contoured on CT scans from 70 HNC patients. The networks were trained and evaluated using five-fold cross-validation and ensemble learning for 60 patients with (1) 3D patch-based UNets, (2) multi-view (MV) voxel classification networks and (3) sequential UNet+MV. The performances were evaluated using Dice similarity coefficients (DSC) for automated and manual segmentations for individual levels, and the planning target volumes were extrapolated from the combined levels I–V and II–IV, both for the cross-validation and for an independent test set of 10 patients. The median DSC were 0.80, 0.66 and 0.82 for UNet, MV and UNet+MV, respectively. Overall, UNet+MV significantly (*p* < 0.0001) outperformed other arrangements and yielded DSC = 0.87, 0.85, 0.86, 0.82, 0.77, 0.77 for the combined and individual level I–V structures, respectively. Both PTVs were also significantly (*p* < 0.0001) more accurate with UNet+MV, with DSC = 0.91 and 0.90, respectively. The accurate segmentation of individual LN levels I–V can be achieved using an ensemble of UNets. UNet+MV can further refine this result.

## 1. Introduction

Head-and-neck cancer (HNC) radiotherapy (RT) planning frequently includes the contouring of neck lymph nodes (LN) as a part of the elective RT target volume. However, the manual delineation of the elective target volume is a labour-intensive task that is prone to inter-observer variation [1], despite the availability of delineation guidelines [2], making automated methods attractive, as an alternative to manual segmentation.

Over the last few years, developments in deep learning approaches have shown impressive results for automated segmentation of organs at risk (OAR) by using convolutional neural networks (CNN) [3,4,5,6] and for pathology detection [7], including the deep learning-based delineation of elective targets such as the combinations of neck LN levels, which has only more recently been investigated [8]. Most studies that demonstrated automated LN segmentation with deep learning, incorporated all LN levels or all of those levels relevant to the primary HNC location in one structure, rather than focusing on individual LN levels [7,8,9,10,11]. The methods that segment multiple lymph levels in one structure, however, are not generalizable to all primary HNC locations and tumour stages and require separate networks for contouring different combinations of lymph node levels. Therefore, it would be desirable to have a more general and flexible approach that concurrently and accurately contours individual LN levels and hence can be used for all HNC patients regardless of the subtype and the specific lymph levels required for RT treatment planning.

The automated segmentation of the LN levels is a challenging task because of anatomical limitations in the manual reference. The guidelines prescribe delineation based on anatomical markers in axial slices and assume that no voxels of levels II, III and IV can exist in the same axial plane, irrespective of the curvature and pitch of the neck. In addition, the LN target volumes do not encompass anatomical structures, but rather the expansions of groups of LNs.

In this work, three combinations of deep learning networks were investigated to segment individual LN levels I–V as separate structures. To do this, we evaluated the performance of two CNNs, alone and in combination. First, since UNet is a widely established CNN that is used for a variety of imaging-related problems [12] and since it was used in two other studies for combined lymph structure segmentation [9,13], we included a patch-based UNet variant as a baseline model configuration. Other works have suggested the use of voxel-classification methods for individual LN level segmentation using a 3D multi-scale network [14], as well as 2.5D (multi-view; MV) networks for several segmentation challenges (multiple sclerosis [15], ocular structures [16], abdominal lymph structures [17], head-and-neck tumors [18]). Because 2.5D networks may more effectively learn features in the presence of little data [19] and because voxel classification may better resolve local ambiguities near level transitions, a multi-view convolutional neural network (MV-CNN) was included as our second configuration. This method, however, appears limited by a systematic over-estimation of foreground classes [18]. Therefore, as our third configuration, UNet was used for foreground segmentation, and subsequently MV was used for classifying the foreground voxels into individual LN levels. This way, the over-estimation of foreground classes seen in MV models was effectively eliminated.

This work expands the existing literature by demonstrating the feasibility of deep learning for auto-segmentation for the target definition of individual LN levels I–V towards a flexible RT planning for locally advanced HNC. Based on earlier work, we estimate that accurate performance levels are attained for the segmentation of individual LN levels I–V with Dice similarity coefficient (DSC) of at least 0.8 [9,13,14] and we hypothesize that the contours can be obtained with such accuracy levels for the majority of patients, using one or more of the proposed deep learning configurations.

## 2. Materials and Methods

### 2.1. Data Acquisition

This retrospective study was exempted from requiring participants’ informed consent by the medical ethics committee and was performed using the three-dimensional (3D) planning computed tomography (CT) scans (GE Discovery 590RT, helically scanned) of 70 patients treated between 2019 and 2022 with (chemo-)radiotherapy for locally advanced HNC, of which 60 were used for training and testing, and 10 were retained for an independent test set. We used isotropic, in-plane acquisition resolutions of [0.92–1.56] mm and a 2.5 mm slice thickness, except for two cases in which the slice thickness was 1.25 mm. The CT acquisition dimensions were 512 × 512 × (147 − 361) voxels. Patient-specific radiotherapy head-and-neck moulds and immobilisation masks were used to position the patients in a neutral position. Ground truth contours were created for the specific purpose of this study by manual contouring of individual LN levels I–V according to contouring guidelines [1], by two experienced radiation oncologists (GJB, MRV). During contouring, levels IV and V are regarded as the combinations of IVa, IVb and Va, Vb, respectively. No HNC disease stages or patients with positive LNs were excluded, provided that they had elective LN levels contoured for at least one side. In patients with only one side contoured, the LN level contours of the side that contained no diagnosed disease were added, such that all patients had all individual levels at both sides contoured.

### 2.2. Pre-Processing

For all patients, planning CTs and structure sets were initially interpolated to the same isotropic 1.25 mm^3^ voxel spacing by 3rd-order and nearest-neighbour interpolation, respectively. This spacing was chosen to minimize image interpolations, whilst making sure the network’s filters were of equal size in each orthogonal plane for all patients.

### 2.3. Experimental Outline

We investigated the performance of three model configurations, i.e., UNet (Figure 1), MV (Figure 1) and UNet+MV (Figure 1). As a baseline reference, we investigated a multi-class, patch-based UNet, which concurrently classifies all lymph levels as separate classes in a single step. This was compared to a per-voxel classification approach that uses an MV-CNN, which is a 2.5D network that uses multiple resolutions of orthogonal views to classify the voxel where the planes cross. In the interest of time, this model used a preconstructed mask to provide the network with the information on which voxels it should consider for segmentation (cyan in Figure 1). Lastly, we investigated a two-step approach, which is essentially a combination of UNet and MV: we used a single-class UNet for segmenting the combined structure of LN levels I–V in an initial step, after which MV was applied only to the detected foreground voxels and classified each voxel in the combined structure into individual lymph levels UNet+MV. Schematic representations of the used UNet and MV networks are displayed in the blue and red boxes in Figure 1 and Figure 2, respectively.

### 2.4. Model Training

Model training, validation and evaluation were performed on four NVIDIA-GeForce GTX 2080 TI graphics processor units (GPUs), a 64 GB RAM system with an Intel^®^ Core™ i9-9900KF CPU @3.6 GHz processor, using the GPU version of TensorFlow (Version 2.2.0) with Cuda 10.1 and Python (Version 3.8.10). The TensorBoard (Version 2.2.2) callback was used for tracking the training and validation scores, whilst only the best model in terms of DSC was saved. The models were trained using the Adam optimizer [20]. All models were trained using standard values in Keras, with an initial learning rate of 0.001, β_1_ = 0.9, β_2_ = 0.999 and ε = 1 × 10^−7^ To reduce the divergence of the model weights at later stages of training, an exponential learning rate decay scheduler was used to decrease the learning rate by 5% with every epoch, up to a minimum of 0.0001. Dropout was switched off at test time. All models were trained using 5-fold cross-validation, with a train\test split of 48\12 cases every fold. To minimize the training variation, we used ensemble learning [9,21,22,23], where the highest cumulated in-class segmentation probability of 5 sequentially trained networks decided the final segmentation map. The training and evaluation times were saved.

#### 2.4.1. UNet

The network that was used is an adaptation of a vanilla UNet [12], where residual blocks were added to reduce the effect from vanishing gradients in deeper layers of the model [24,25], similar to those used by Millerari et al. [26] Batch normalization was performed after every (3 × 3 × 3) 3D convolution, before the non-linear activation function. We used patch-based training of the 3D UNet to ensure the network fitted on our video card [27,28]. During training, patches of 64 × 64 × 64 voxels were sampled randomly from two pre-defined, unilateral regions of interest (ROI) of 280 × 200 × 280 mm^3^ in volume that were known to contain the combined structure of LN levels I–V for every patient on each side. Binary and multi-class dice loss functions were used for optimization. The multi-class DSC loss was defined as the sum of individual foreground class losses (Equation (1)):(1)$$DSC_{loss}=\overset{M}{\underset{m=1}{\sum }}W_{m}\cdot DL_{m}$$

Here, *W_m_* are the class weights that are calculated using the Python’s scikit-learn module [29], *m* ranges from 1 to *M* and denotes class indices, where *M* is the number of classes. *DL* is the *DSC* loss, defined as 1 minus the *DSC* score (Equation (2)):(2)$$DL_{m}=1-\frac{2\cdot \left|A_{m}\cap B_{m}\right|}{\left|A_{m}\right|+\left|B_{m}\right|}$$

Here, *A_m_* and *B_m_* denote the predicted and manual reference binary sets of class *m*, respectively. In the case of binary segmentation, *DSC_loss_* is reduced to the latter loss function. For patches that contain a limited amount of foreground voxels, *DSC_loss_* becomes ill-defined (the denominator in *DL_m_* is not constrained to values larger than 0). To ameliorate this, we used a Gaussian sampling method, where the mean and standard deviation of the x, y and z coordinates are calculated from the centre of mass of the combined, binary structure of LN levels I–V of all patients. Subsequently, we used a truncated normal distribution to sample patches, such that they were constrained to be entirely within the region of interest. The weights were initialized using the standard initialization method in Keras (glorot uniform initialization). The models were optimized for 100 epochs. However, it should be noted that the use of an epoch in a patch-based setting is arbitrary, because patch sampling is perfrmed at random, and thus a different sub-set of all data is seen by the network in each epoch. The number of training pairs seen by the network per epoch was set to 4096, which corresponded to roughly 34 training patches per side per patient.

#### 2.4.2. Multi-View

MV-CNN is a voxel-wise classification method, for which we predefined which voxels to classify. For C2, this information was provided by a pre-constructed mask, indicated in cyan in Figure 1, which was constructed by a uniform expansion of the manual reference by a margin of 15 mm. This margin was chosen as a balance such that no foreground voxels would be segmented at the border of this mask, while also minimizing the training and evaluation times. In contrast, for C3, the pre-constructed mask was determined by the foreground segmentation result of UNet. Our multi-view network was adapted from a previous classification study [16]. Batch normalization was applied after every (3 × 3) 2D convolution layer, before the non-linear activation function. Three context pyramid scales, 0, 1 and 2, were included to incorporate multi-view information from 4, 8 and 16 cm around the query voxel, respectively. This was done by sampling every, every other and every fourth voxel for scales 0, 1 and 2, respectively, for each view. Fewer pyramid scales yielded inferior results, and more pyramid scales would cause the field of view to fall far outside the ROI. The loss function used for voxel classification was categorical cross-entropy (CCE; Equation (3)):(3)$$H\left(p,q\right)=-\overset{M}{\underset{m=1}{\sum }}\;\overset{A}{\underset{a=1}{\sum }}p\left(a,m\right)\;\log\left(q\left(a,m\right)\right)$$
where *p*(*a*,*m*) represents a reference distribution of *a* ∈ *A*, given by the manual annotations, *q*(*a*,*m*) is a query distribution, *A* is a set of observations, *m* denotes class indices and ranges from 1 to *M*, and *M* is the number of classes. The network was optimized for 1000 epochs (batch size = 32). In every epoch, a different random sub-set of at maximum 20% of all training pairs was sampled to allow for varied training and validation. Random over-sampling of minority classes was applied to reduce the effects of class imbalance.

#### 2.4.3. Data Augmentation

Data augmentations were the same for all models and were performed on the fly. Augmentation involved random flipping, rotation and contrast adaptation, with chances of each augmentation occurring being 50%, 40% and 40%, respectively. Flipping was carried out in the left–right direction. Rotation was applied in either the sagittal or the transversal plane, with an angle that was uniformly sampled from [−5, +5 degrees]. Rotated images were acquired by 3rd order spline interpolation for the CT image and by nearest-neighbour interpolation for the corresponding segmentation maps. The default window level center (C_C_) [width (C_W_)] was 0 [700], as was previously used for lymph structure segmentation [9]. If contrast adaptation was applied, alternative window level center, and width were sampled from normal distributions, with μ_C_ = 0; σ_C_ = 3% × 700 and μ_W_ = 700; σ_W_ = 3% × 700, respectively.

### 2.5. Post-Processing

In all segmentation maps, the combined structure of LN levels I–V was post-processed with hole filling and by subsequently removing all but the largest connected components. To investigate the agreement in the resulting planning target volumes (PTV), the resulting segmentations of combined structures of LN levels I–V and II-IV were expanded by a margin of 4 mm and were denoted as PI–PV and PII–PIV. These two PTVs were chosen because they were used for planning the majority of HNC sub-types.

### 2.6. Evaluation and Statistical Analysis

The evaluation of a full 3D image by UNet was achieved by sliding the 64 × 64 × 64 UNet field of view over the image with stride 32 and subsequently only evaluating the central 32 × 32 × 32 voxels. By doing this, we ensured that the network had sufficient context for reliable inferences, while also making sure that each voxel was classified exactly once. The spatial performance of all models was measured by using DSC, Hausdorff distance (HD) and mean surface distance (MSD) between predictions and manual contours and between the PTVs that resulted from the predictions and manual contours. Because the measures were not normally distributed upon histogram inspection and omnibus test of normality [30], the differences in spatial performance were evaluated by a two-sided Wilcoxon signed-rank test. Bonferroni correction was applied for each model and spatial metric separately to account for multiple comparisons. The volumetric agreement was assessed with intra-class correlation [31] (ICC; two-way mixed effects, single measurement, consistency) coefficients and volume outside of the manual contour. Finally, cases with a median DSC in the lowest quartile of the UNet+MV configuration were qualitatively reviewed by GJB. Cases from each quartile (Q1–Q3), as well as several informative examples, were chosen for display, such as one patient who underwent laryngectomy surgery. This case was included during training to maximize the number of training samples but was omitted from the calculations of the model performance metrics, because the anatomical landmarks normally required for manual contouring were not present in this patient’s anatomy.

### 2.7. Independent Validation

To assess the model generalizability, the two best performing models (UNet and UNet+MV) were tested on the independent test set of 10 patients. These were unique samples that were not seen or used during the model development. For this independent testing, the UNet and UNet+MV models were re-trained using the complete cross-validation dataset (60 patients) as the training data. All training and evaluation settings were identical to the cross-validation setting.

## 3. Results

In the cross-validation set, the mean age ± standard deviation was 64.0 ± 10.4 (*N* = 49) and 58.5 ± 4.9 (*N* = 11) for males and females, respectively. UNet and UNet+MV showed better agreement with the manual reference than MV for the complete LN structure, all individual LN levels and both PTVs (Table 1). UNet+MV typically showed the better segmentation performance of the combined LN structure, individual levels II–IV and both PTV structures (Figure 3, Figure 4, Figure 5 and Figure 6; Table 1). In addition, UNet+MV showed the highest volumetric agreement with the manual reference for all structures (Figure 4). Overall, UNet+MV significantly (*p* < 0.0001) outperformed the other models, with the DSCs (median [interquartile range (Q1–Q3)]) of all individual LN structures present in the dataset being 0.804 [0.763–0.814], 0.658 [0.616–0.678] and 0.821 [0.769–0.831] for the models UNet, MV and UNet+MV, respectively. Even with some deformation, e.g., patient not aligned straight in the mask, median-level DSC results were attained (e.g., Figure 3, second column). MV often (Figure 3, Ax. 1 and Cor. 1 rows) overestimated the segmented combined LN volume medially.

UNet+MV showed significantly higher DSCs for the complete LN level I–V structure, individual levels II–IV and both PTV structures (Figure 5, *p*-values in figure). However, UNet showed higher spatial agreement with the manual reference for LN level I. The transitions of LN levels II–III by UNet+MV typically agreed most strongly with the manual reference. All models commonly disagreed with the manual reference on the caudal and cranial ends of LN level V. In addition, there existed a substantial disagreement on the lateral and dorsal ends of this structure in the model predictions. The models benefitted marginally from ensembling all model configurations for all classes, as the results from the model ensembles were more consistent (Table 1). The models were optimized for a median [range] of 10.3 [9.6–10.9] h, except for MV–only, which was optimized for 19.8 [18.7–21.2] h. The inference time for all UNet models was 2.1 [1.8–2.4] minutes, whereas the MV inference time, which is proportional to the size of the input mask, was 6.0 [5.4–7.0] and 1.0 [0.8–1.3] minutes per patient for the MV–only and UNet+MV configurations, respectively.

By visually comparing the model and manual reference contour pairs in the worst-performing quartile (*N* = 15), several trends were observed. First, the manual reference was judged to be suboptimal (i.e., not according to the contouring guidelines; Figure 6A–E) for at least one level in 6/15 cases. In these six patients, one, four, two, one and three inaccuracies were found in each respective LN level I–V. Second, the level II–III transitions predicted by UNet+MV were typically more accurate than those obtained from the manual reference, and UNet+MV also often outperformed UNet at this transition (Figure 6F–I). Third, the predictions of LN level II by UNet and UNet+MV were visually more accurate than those of the manual reference at the cranial limit (Figure 6J). Fourth, the automated methods showed a large variation in disagreement with the manual reference for LN level V (Figure 6A,E,H,I,L). In cases where the automated methods showed considerable disagreement with the manual reference, pitch, rotation and/or tilt were often underlying confounders (Figure 6K–M), especially for LN level V (Figure 6M), or there were anatomical variations such as malnourishment (Figure 6F) and laryngectomy (with fewer anatomical landmarks available; Figure 6N–O)). Cases with a coronal tilt showed disagreement in contralateral structures of the same level (Figure 6M). Among cases of the first quartile, there were no particularities in the manual reference.

In the independent test, the mean age ± standard deviation was 66.3 ± 10.1 (*N* = 7) and 64.3 ± 13.6 (*N* = 3) years for males and females, respectively. The median [interquartile range (Q1–Q3)] DSCs of all individual LN level structures were 0.769 [0.703–0.834] and 0.809 [0.729–0.852] by the UNet and UNet+MV configurations, respectively, and differed significantly (*p* < 0.0001). UNet+MV showed significantly higher DSCs for the complete I–V structure, LN levels II and III, as well as both extrapolated PTVs (Table 1; Figure 7). For reference, volumetric performances of the independent test set are included in Appendix A.

## 4. Discussion

Our results suggest that accurate contours of individual LN levels I–V can be obtained using UNet (complete I–V structure median DSC = 0.859; individual structure DSC = 0.804), and that these results can be further refined by using a UNet+MV sequential model (complete I–V structure DSC = 0.866; individual structure DSC = 0.821). Despite a limited gain compared to UNet, UNet+MV exhibited a significantly better spatial performance for the complete I–V structure, individual levels II–IV and both PTV structures, and better volumetric performance for all structures. Comparable results were achieved using an independent test set for the model configurations UNet and UNet+MV, suggesting that the models have the ability to generalize beyond the data used for model training and development.

These results, however, should be interpreted with some care. A review of patients with a median DSC in the lowest quartile (*N* = 15) highlighted cases where the automated methods were factually closer to the truth than the manual reference, due to inconsistencies in the manual reference that arose from patient angulation and anatomical limitations in contouring guidelines (Figure 6A–M). In addition, all models were considerably less accurate for levels IV and V. Several factors may have contributed to this. First, it is known that DSC is dependent on the structure size [32]; therefore, the small volumes of the levels IV–V likely negatively influenced DSC, which was especially true for malnourished patients (Figure 6M). Such a case was observed in the independent test, where LN level V had a manual reference volume below the typical range (5 mL) and was almost completely missed (Figure 7; Appendix A-LN level V). Second, despite the measures that were taken to prevent most patient angulation during scanning, considerable patient angulation was sometimes seen. This could be due to anatomical variations and to some patients’ inability to lie with their head down. This may also have contributed to a larger variation in the manual reference and may have led to disagreements between the predictions and the manual reference. This problem has recently been addressed in another study by Weissmann et al. [13]. Because the contouring guidelines do not take into account the curvature of the neck and the patient’s pitch, tilt and rotation, it can be argued that the predictions may be more factually “correct” than the manual reference when this is the case. Alternatively, if the goal of DL methods is to emulate the contouring guidelines, the networks could be trained using explicit information of slice orientation. Variations in slice plane orientation are especially problematic for level V, because, for example, the lower axial end of this structure contour in the manual reference is defined by the “plane just below the transverse cervical vessel” [2,33]. This caused larger inconsistencies in the manual reference for LN-level V highly pitched patients, compared to patients with other levels. The same holds for the contralateral structures from the same level for patients with a coronal tilt. The current guidelines prescribe level contours of both sides starting at the same axial slice clinically, even though the tilt leads to different predictions for either side for the automated methods. Similarly, although predictions generally show disagreement in the caudal end of level IV and both axial ends and dorsal borders of level V, it should not be concluded that predictions are inaccurate for these regions. Rather, the way that the contouring guidelines were set up can cause peculiarities for patients with large pitch and/or tilt when comparing with more standardized, automated methods. Although it could seem like the rational step to take, it is not a given fact that redefining contouring guidelines to be less dependent on anatomical landmarks in a certain slice and patient angulation would be better for the clinical practice. Such guidelines would be more labour-intense for the clinician, which will need to consider more strongly the 3D information of the patient. However, such an approach may result in more accurate data, which in the long run, will be more informative to the network and result in more consistent contours.

To put the results of this study into perspective, we compared our results to others in the relevant literature on automated lymph level segmentation of combined lymph levels, which reported a mean DSC range of 0.64–0.82 [34] Commercially available contouring software (Limbus Contour build 1.0.22) was evaluated for the neck lymph nodal structures [11], but it was reported that the performance could still be improved (mean DSC = 0.75). Cardenas et al. reported an accurate segmentation performance of the combined LN level I–V and II-IV clinical target volumes (CTV; both DSC = 0.90) [9], but it should be noted that an inspection of example segmentations suggested that these structures more closely resembled PTV structures from our institute. We believe that our finding of PTV overlap of UNet and UNet+MV (PTV I–V and II–IV DSCs = 0.91, 0.90, respectively) is in line with, if not better than, the segmented structures reported by Cardenas et al. To the best of our knowledge, the work of Van der Veen et al. [14] was the first to involve the automated segmentation of individual levels I and V and reported segmentation accuracies (without expert intervention) of DSC = 0.73, 0.61 and 0.79 for levels I and V and the combined II–IV structure, respectively. Interestingly, however, these results seem to more closely resemble the results obtained with our second configuration (level I, V DSCs = 0.70, 0.61, respectively). This is not unexpected, because the MV configuration involves a direct voxel classification method that uses multiple scales, similar to the proposed method by Van der Veen et al., but differs in the 2.5D convolution kernel, whereas Van der Veen et al. used a fully 3D kernel.

The model application times are sufficient for clinical use, but can still be improved. Typical whole-image full segmentation by UNet takes time in the order of seconds, but since this UNet was trained in a patch-based fashion, it required application to all parts of the image, such that each part of the image was seen by the 32 × 32 × 32 center patch exactly once. This procedure was not optimized for speed and could likely still be accelerated considerably. Similarly, the MV models were not optimized for speed. For example, when processing neighbouring voxels, there existed much overlap between the extracted patches, even though each patch was extracted separately in the current implementation.

Our research has some limitations. First, we only indirectly investigated the implications of model predictions for RT treatment planning by investigating the overlap of the two predicted PTVs with the manual reference. Future work may investigate whether the predicted volumes lead to improved dose–volume histograms in OARs and target volumes when using them in a treatment planning system. Second, we did not include LN levels VI and VII because these are less frequently clinically used. Since these are central levels and require a larger region of interest to be considered for learning, deep learning frameworks aiming to include these structures may focus on patch-based training with sampling from both sides simultaneously or by defining two left/right and one central ROI.

## 5. Conclusions

We demonstrated that a UNet can accurately (DSC > 0.8) segment individual LN levels I–V for the majority of patients and that this result can be further refined by using a UNet for the segmentation of foreground structures, followed by a sequential voxel classification network. With this generalized approach, any set of lymph levels can be combined to define patient-specific LN level target structures. When dealing with angulated patients, one should be aware that the current contouring guidelines can lead to situations where the LN level contours may become inconsistent, which may be prevented by using more standardized, automated deep learning methods. Future work should investigate whether clinically acceptable RT plans can be obtained using predicted contours.

## Figures and Tables

**Figure 1 cancers-14-05501-f001:**
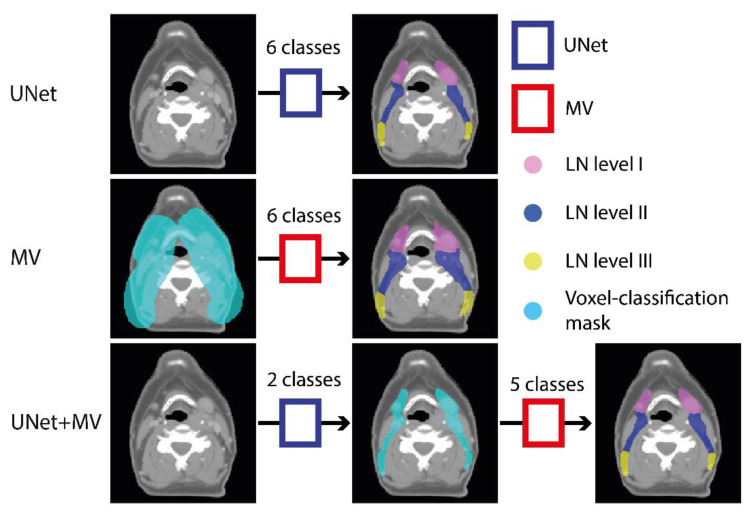
Schematic overview of the experimental outline. UNet (blue boxes) and MV (red boxes) were used to make three model configurations. In the first configuration, a patch-based UNet segments the background and LN levels I–V directly from the planning CT. In the second configuration, MV classifies the background and LN levels I–V voxels from within a preconstructed mask (cyan). In UNet+MV, a patch-based UNet first segments the combined structure of LN levels I–V. This is subsequently used as a mask (cyan) for MV to subsequently classify positive voxels into individual levels I–V. The details of both models are given in Figure 2. Abbreviations: MV: multi-view; CT: computed tomography.(Also shows in Appendix A).

**Figure 2 cancers-14-05501-f002:**
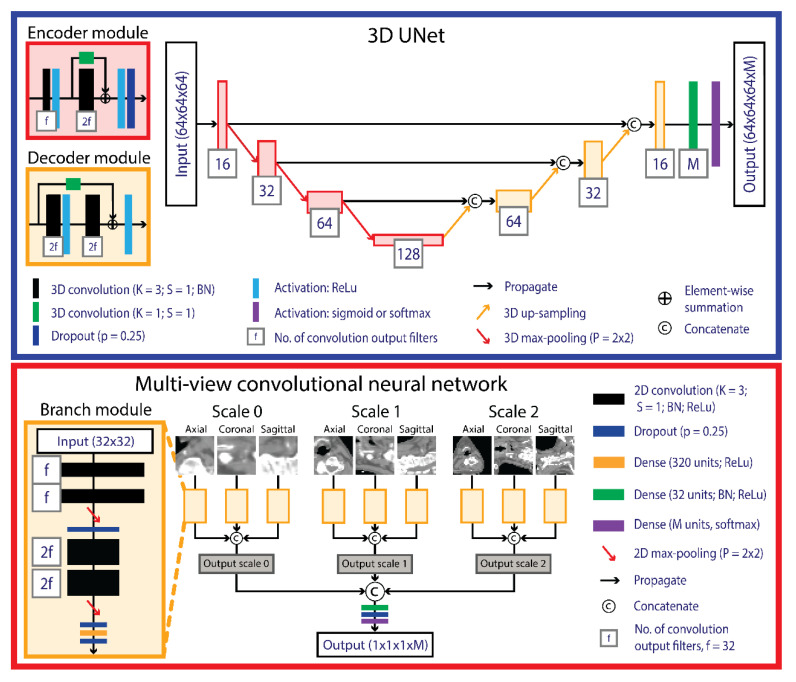
Schematic overview of the UNet (blue box) and MV (red box) networks. UNet consists of an encoder (**left**) and a decoder (**right**) pathway that generates binary segmentation maps from 64 cubed voxel patches sampled from planning CTs. MV uses three multi-view branches that build up to each anatomical plane within a scale block, the output of which is concatenated and used as the input for the multi-scale branched architecture. The thickness of the convolutional blocks corresponded with the number of filters used. The number of output classes (M) was six for UNet in the UNet-only configuration and two for UNet in the UNet+MV configuration. M was six for MV in the MV-only configuration and five in the UNet+MV configuration. Abbreviations: MV: multi-view; ch: number of channels; BN: batch normalization; ReLu: rectified linear unit; f: number of output filters; M: number of output classes; K: convolution kernel size; S: convolution stride; BN: batch normalization; p: dropout fraction: CT: computed tomography.

**Figure 3 cancers-14-05501-f003:**
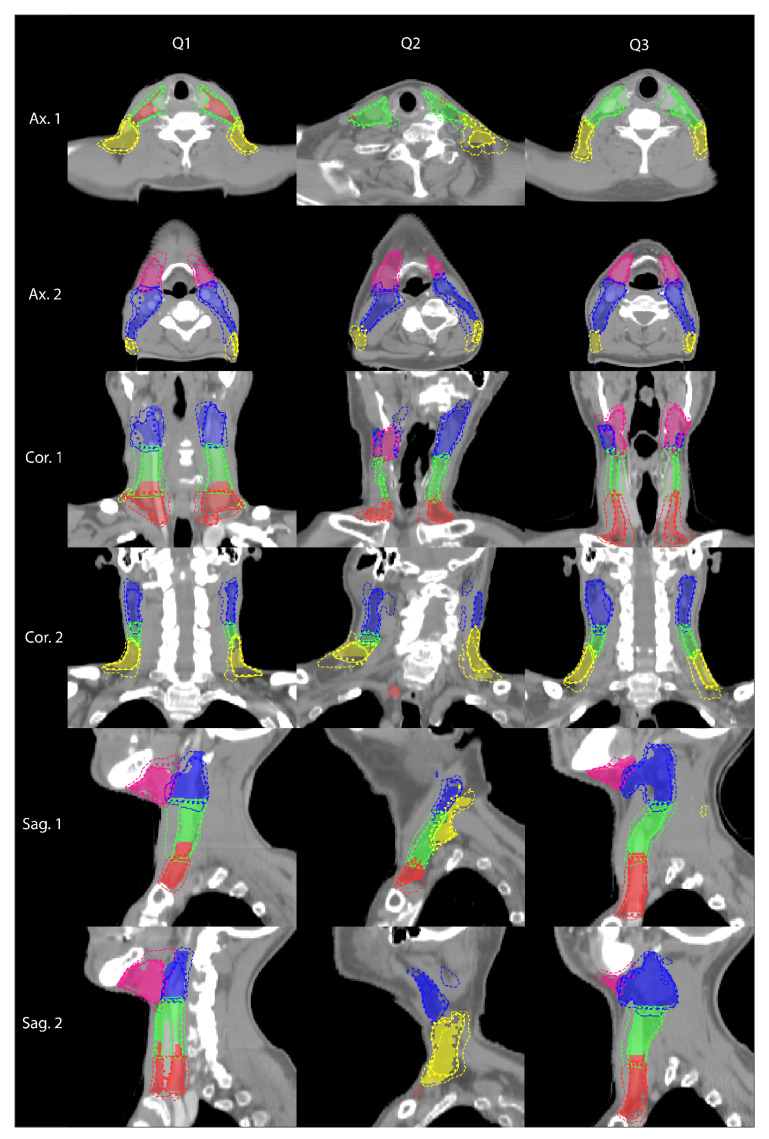
Example segmentations selected from the first (Q1), second (Q2) and third (Q3) quartile in terms of DSC averaged over individual LN levels I–V. The filled region is the manual reference. The solid, dashed and dotted lines correspond to the predictions of the model configurations of UNet, MV and UNet+MV, respectively. LN levels I–V are indicated in pink, blue, green, red and yellow, respectively. The low average DSC in Q1 was in part attributed to an error in the manual reference level III–IV transition. Abbreviations: DSC: dice similarity coefficient; LN: lymph node.

**Figure 4 cancers-14-05501-f004:**
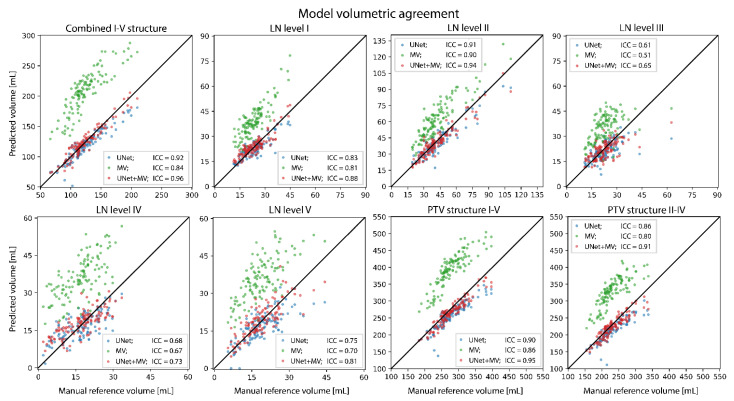
Predicted and manual reference volumes for all structures. Abbreviations: ICC: intra-class correlation (two-way mixed, single measures, consistency).

**Figure 5 cancers-14-05501-f005:**
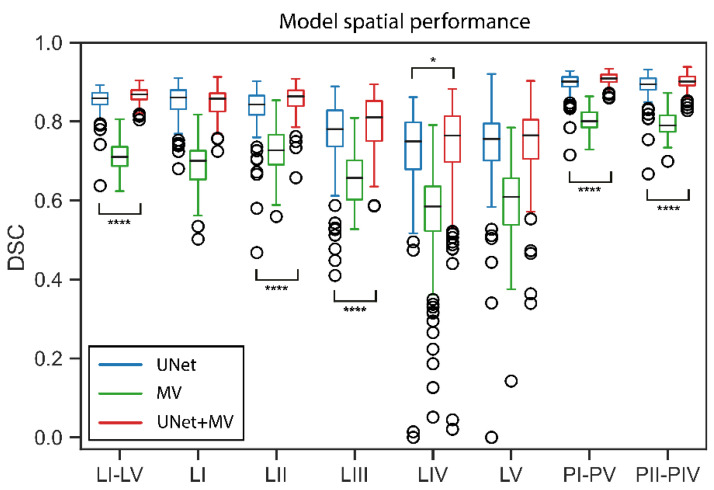
Spatial performances of UNet, MV and UNet+MV model configurations for DSC, HD and MSD measures. Statistical significance marking of the MV configuration was omitted because differences between MV and other model configurations were always significant. Structures for which differences between UNet and UNet+MV were statistically significant are denoted by significance bars. *: *p* < 0.05; **: *p* < 0.01; ***: *p* < 0.001; ****: *p* < 0.0001; Abbreviations: DSC: dice similarity coefficient; MV: multi-view; HD: Hausdorff distance; MSD: mean surface distance.

**Figure 6 cancers-14-05501-f006:**
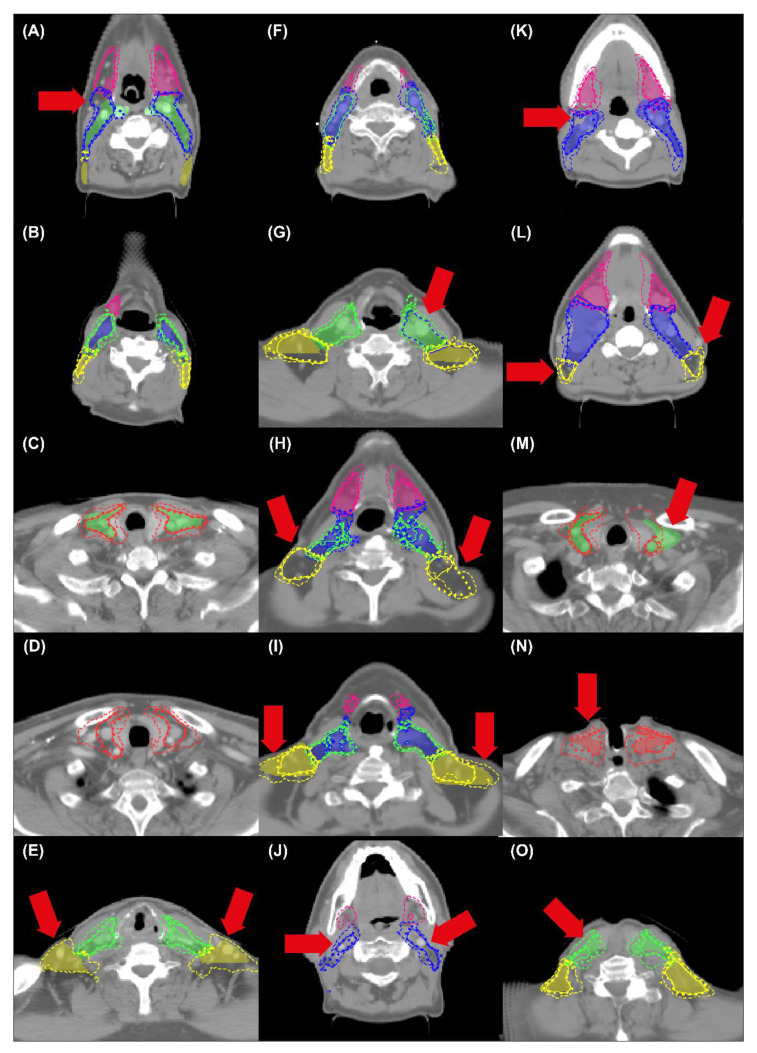
Examples from the worst-performing quartile samples in terms of DSC averaged over individual LN levels I–V. The filled region is the manual reference. The solid, dashed and dotted lines correspond to the predictions of the UNet, MV and UNet+MV model configurations, respectively. LN levels I–V are indicated in pink, blue, green, red and yellow, respectively. Arrows indicate specific locations of interest. Abbreviations: DSC: dice similarity coefficient; LN: lymph node.

**Figure 7 cancers-14-05501-f007:**
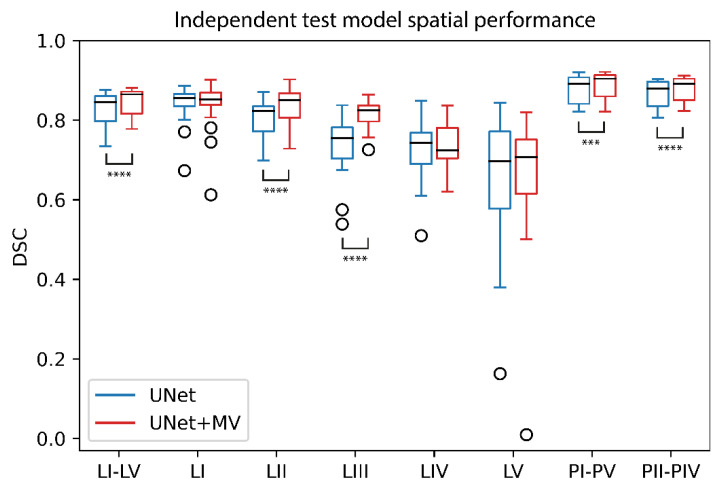
UNet and UNet+MV spatial model performances in the independent test. Structures for which differences between model configurations were statistically significant are denoted by significance bars. ***: *p* < 0.001; ****: *p* < 0.0001; Abbreviations: DSC: Dice similarity coefficient; MV: multi-view.

**Table 1 cancers-14-05501-t001:** The reported values denote the range of median DSCs produced by five individual models and ensemble model combinations of UNet, MV and UNet+MV configurations after post-processing. Ensemble results that showed higher spatial agreement than the most accurate individual model are denoted in bold. Ensembles increased result consistency and typically outperformed any of the standalone models for all configurations. Abbreviations: MV: multi-view; Ind. individual; Ens: ensemble; LN: lymph node.

	Cross-Validation						Independent Test	
	UNet		MV		UNet+MV		UNet	UNet+MV
	Ind.	Ens.	Ind.	Ens.	Ind.	Ens.	Ens.	Ens.
LN I–V	[0.850–0.852]	**0.857**	[0.692–0.706]	**0.708**	[0.860–0.862]	**0.867**	0.846	0.865
LN I	[0.849–0.855]	**0.860**	[0.682–0.695]	**0.700**	[0.851–0.856]	**0.857**	0.856	0.852
LN II	[0.827–0.834]	**0.840**	[0.702–0.720]	**0.726**	[0.856–0.858]	**0.862**	0.824	0.850
LN III	[0.771–0.781]	0.781	[0.628–0.653]	**0.656**	[0.802–0.812]	0.810	0.755	0.825
LN IV	[0.714–0.746]	**0.748**	[0.559–0.585]	0.583	[0.757–0.764]	0.764	0.743	0.724
LN V	[0.738–0.751]	**0.754**	[0.572–0.604]	**0.610**	[0.753–0.761]	**0.763**	0.697	0.707
PI–PV	[0.897–0.898]	**0.899**	[0.779–0.788]	**0.798**	[0.899–0.900]	**0.908**	0.892	0.904
PII–PIV	[0.887–0.891]	**0.892**	[0.768–0.782]	**0.788**	[0.899–0.900]	**0.902**	0.893	0.892

## Data Availability

The data used in this study is not made publically available for reasons of consent of the participants.

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
