# Peer review of "Deep Learning for Automated Elective Lymph Node Level Segmentation for Head and Neck Cancer Radiotherapy"

_cancers, 2022, doi:10.3390/cancers14225501_

Round 1

Reviewer 1 Report (Previous Reviewer 1)

Thank you for addressing the comments. I have no further comments.

Author Response

No further changes were requested by Reviewer 1.

Reviewer 2 Report (New Reviewer)

Authors have evaluated three deep learning approaches for automated elective lymph node level segmentation for head and neck cancer radiotherapy.

1-     The abstract should be a single paragraph and should follow the style of structured abstracts, but without headings.

2-     There are different font type in line [36-37].

3-     It would better if the authors include a table of participant demographics/characteristics. Minimum information should enough for e.g. (include number and sex of participants, mean age ± standard deviation (SD) or median age). An example of patient demographics https://doi.org/10.1007/s00247-022-05510-8.

4-     Some new references should be added to improve the literature review—for example, https://doi.org/10.1016/j.compbiomed.2022.105295; https://doi.org/10.3390/app12115500.

Author Response

We thank the reviewer for bringing the written problems to our attention. We will address them point-by-point here.

1. The abstract should be a single paragraph and should follow the style of structured abstract, but without headings.

  • The paragraph headings were removed

2. There are different font types in line [36-37]

  • The different font type was removed and change to palatino linotype

3. It would be better if the authors included a table of participant demographics.

  • We included textual information regarding the mean age +/- standard deviation for males, females in the cross-validation and test sets, separately. This includes a distribution of the part male/female in both data sets.

4. Some new references should be added to improve the literature review.

  • We included the references requested by the reviewer and an additional recent reference about LN level segmentation to the literature review in the introduction and discussion sections.

This manuscript is a resubmission of an earlier submission. The following is a list of the peer review reports and author responses from that submission.

Round 1

Reviewer 1 Report

A very nice and complete study presented in this manuscript. The ML techniques are very interesting and the goal to auto-contour individual lymph node levels makes it highly relevant for the field. 
The manuscript is well written with clear description of the methods and results.

My only question is whether you included lymph node positive patients in the dataset, this is not clear from the manuscript. If so, please report the influence of positive nodes on the accuracy of that specific level. If not, please elaborate in the discussion why not and how your results could be used for node positive patients.